# The Effects of Risk Perceptions Related to Particulate Matter on Outdoor Activity Satisfaction in South Korea

**DOI:** 10.3390/ijerph17051613

**Published:** 2020-03-02

**Authors:** Bomi Kim, Eun Joo Yoon, Songyi Kim, Dong Kun Lee

**Affiliations:** 1Interdisciplinary Program in Landscape Architecture, Seoul National University, Seoul 08826, Korea; adees@snu.ac.kr; 2Korea Research Institute for Human Settlements, Sejong 30147 Korea; yoonej@krihs.re.kr; 3Tourism Policy Research Division, Korea Culture and Tourism Institute, Seoul 08826, Korea; 4Department of Landscape Architecture and Rural System Engineering, Research Institute of Agriculture and Life Sciences, Seoul National University, Seoul 08826, Korea

**Keywords:** air pollution, outdoor activities satisfaction scale (OASS), PM_10_, PM_2.5_, South Korea, structural equation model (SEM)

## Abstract

In recent years, the Korean public has become aware of the form of air pollution known as particulate matter, with a consequent growth of public anxiety causing a negative risk perception about outdoor activity. This study aims at determining the causal relationship between risk perceptions about particulate matter and outdoor activity satisfaction in South Korea. An Internet survey was conducted with 412 people, and a structural equation model was used to perform confirmatory factor analysis. The statistically significant results show that the perceived risk of particulate matter is higher when people do not show interest in or trust public opinion or policy on the subject. This increases people’s perceptions of health risks, which in turn lowers their satisfaction with outdoor activity. Although trust levels in public opinion or policy had a positive impact on outdoor activity satisfaction, this was not statistically significant. These results are expected to contribute to risk communication guidelines in public opinion reporting and to the direction of environmental health policies in developing countries with high levels of air pollution, such as particulate matter.

## 1. Introduction

The situation regarding particulate matter (PM), a common indicator of the severity of air pollution, has become a serious problem for several developing countries [1,2,3]. South Korea is likewise in a critical situation. In particular, its PM2.5 concentration (24.8 μg/m^3^/year) ranks first among the OECD countries in 2019. Sixty-one of South Korea’s cities were included among the top 100 cities with high pollution levels, the highest number among member countries [4].

PM in South Korea has unique seasonality originating from geographic characteristics, climate, and seasonal wind direction. From late autumn to spring, high concentration levels can be measured that are affected by neighborhood countries or domestic conditions (e.g., yellow sand, diesel cars, coal powered energy, and so on) [5,6,7,8,9]. However, in summer, seasonal variation shows little effect on PM. Therefore, Korean people perceive PM problems differently from spring to winter [10].

In fact, South Korea’s interest in air quality began to rise with the Seoul Olympics, when the government focused on the air pollution situation and improvement measures [11]. In 2001, South Korea established the interim-target level 1 from the World Health Organization (WHO) in consideration of domestic economic and social conditions (Table 1). PM was first measured as total suspended particles (TSP), and in 2001, that standard was removed, and PM10 (1993) and PM2.5 (2011) standards were added. Since 2003, South Korea has established emission standards for emission facilities and diesel vehicles in order to manage the air pollutants that are actually released into the atmosphere. In response to these policies, air quality standards were raised to WHO’s interim-target level 2 in 2007. In the 2000s, air management policies in South Korea were driven by fuel policies. As a result, the annual average concentration of PM10 in Seoul improved from 71 μg/m^3^ (2001) to 49 μg/m^3^ (2010) [10].

After this improvement, PM reduction policies did not receive much attention. However, since the International Agency for Research on Cancer (IARC) of WHO announced PM2.5 as a group 1 carcinogen, PM health risks became a social issue in 2013 [12,13,14]. In 2014, the government introduced Korea’s Fine Dust Forecasting and Alarming System (FDFAS) to reduce the public’s concerns about PM [15]. Since then, the government has sent text messages warning the public of high-concentrations of PM in real time [16]. This triggered the nationwide spread of PM issues, which had been limited to metropolitan areas [15,17]. As negative public opinion related to PM spread, the government proposed various PM reduction strategies over a short time: the “Special Act on the Reduction and Management of Fine Dust” (2016), the “Fine Dust Management Comprehensive Measures” (2017), and the “Fine Dust Reinforcement Measures” (2018) [18]. However, as days of high-concentration PM have increased [10], the period of visual recognition about PM has also increased. In addition, people are more aware of PM risk with the increase in media coverage and warning messages that recommend masks and refrain from outdoor activity (OA) [11,17].

People’s demands for improving air quality have thus increased [12,13]. Accordingly, in 2018, the government strengthened the PM2.5 standard to WHO’s interim-target level three and provided guidelines for strengthening the watch and warning levels without resolving major issues related to PM [17,19]. Although the government is strengthening environmental standards and responding to PM problems, the public’s anxiety about PM has increased [15]. In the end, the PM risk perception (PMRP), which was formed in a relatively short time, affected the lives of the public until this situation was included as a social disaster in the “Disaster and Safety Management Act” (2019) [12,13]. Comprehensively, PM, a previously somewhat unfamiliar issue in South Korea, has become known as a health risk [20], and the growth in public anxiety is causing a negative risk perception about OA that is directly related to health [1,2].

OA is limited by air pollution levels, such as PM [21,22]. Restrictions on OA may lead to increased economic costs because of the use of defensive measures (e.g., masks and air filters) [23] or other alternative measures (e.g., yoga and weight training) to traditional OA [15]. The inconvenience and economic burden of daily life cause concern and influence public opinion about PM, and these concerns are presented as a major issue in the media. In South Korea, where the Internet and social networking services are well-developed, PMRP may be formed depending on the frequency of media coverage, regardless of the actual risk or frequency of occurrence [5]. In addition, as public opinion about PM is worsening, the government is losing credibility with the public due to high-profile and short-term PM policies. It means that the effectiveness and utility of PM policies encounter many problems, which in turn raise the public’s concerns, leading to a vicious cycle of policymaking and implementation where no solution is proposed [17]. Therefore, PMRP is a complex phenomenon consisting not only of actual health risks but also of the social influence of public opinion and frequent policy changes.

However, little research has been conducted on the influence of PMRP on outdoor activity satisfaction (OAS). The main studies on OAS have focused on social exchange, physical strength improvement, and stress relief, which are positive effects of traditional OA [24,25]. Further, they have analyzed satisfaction with outdoor recreation based on demographic characteristics (e.g., gender, age, and education) [1,26]. Although studies on the negative effects of OA have been conducted [27,28,29], only a few of them have focused on the satisfaction experienced during OA as it relates to PM. Hence, the perceived health risks regarding PM in OA are still being debated. In fact, the subjective risk perception of the public is an important factor for understanding how risks exist in the social environment [30] and can explain more clearly the social phenomenon of perceived risk [31]. It is important to understand the relationship between the public’s risk perception and experienced satisfaction under high PM concentration, as it can provide information to stakeholders on how to resolve concerns about the health risks of PM [32,33]. Therefore, the purpose of this study is to investigate the causality between PMRP and experienced OAS of Koreans. The findings are expected to serve for future risk communication guidelines in public opinion reporting and in the direction of environmental health policies in developing countries with severe air pollution.

### 1.1. Theoretical Foundation and Study Model

#### 1.1.1. Risk Perception Variables Related to Particulate Matter

PM is a hazardous air pollutant in urban areas [10,21]. In South Korea, PM is divided into PM10 (smaller than 10 μm) and PM2.5 (smaller than 2.5 μm), based on aerodynamic diameter [10,16]. A smaller aerodynamic diameter of PM has a more direct impact on health and the environment [3,34] and is related to many health risks such as respiratory and lung diseases, thus starting to form the public’s risk perception in South Korea [12,13]. The PMRP can be redefined as a negative perception that is formed by the direct risks caused by PM or the indirect influences of the public, such as public opinion or policies (POP) related to PM.

What are the factors that affect people’s risk perception related to PM in South Korea? PMRP is a complex aggregation influenced not only by health risks but also by social influences, such as POP [5,6]. To select variables related to PMRP, 12 factors, such as PM definition, health symptoms, OA, and countermeasures based on the research of Kim and Lee (2019) [15] were chosen. Thereafter, PMRP questionnaires were constructed. The 12 factors were divided into four clusters through principal component analysis (PCA), but one cluster that was not reliable was excluded. The excluded cluster consisted of two cognitive questions on the level of PM information exposure and PM policies perception. Finally, three clusters consisting of 10 variables were determined as “interest level in POP”, “trust level in POP”, and “perception level of health risk” about PM (Table 3).

In South Korea, where the Internet is well-developed, unlike in other developing countries, there is a unique level of PMRP. South Korea’s PMRPs are formed by POP developed over a short time. Since 2014, public opinion about PM has been excessively influenced by the Internet use and media coverage [20,35,36], and various PM policies have changed to reflect these circumstances [17,19]. These results influenced PMRP. In other words, PM health risk perception can vary depending on one’s interest and surrounding influences about PM. Therefore, this study established the following two hypotheses based on previous studies:

 **Hypothesis 1 (H1).** 
*The level of health risk perceptions about PM has a negative effect on the level of interest in POP about PM.*


 **Hypothesis 2 (H2).** 
*The level of health risk perceptions about PM has a positive effect on the level of trust in POP about PM.*


#### 1.1.2. Outdoor Activity Satisfaction Variables Related to Particulate Matter

People’s positive or negative perceptions of the surrounding environment affect their OA [37,38,39]. People’s OA is expressed as satisfaction or dissatisfaction after their behaviors [40,41,42]. Satisfaction is a term used to describe the feelings experienced after a particular situation. The Likert scale is used as a tool to evaluate respondents’ states of mind [43]. OAS is defined as the degree of satisfaction that results from OA as a part of human behavior. Therefore, it can be expressed as pleasure or satisfaction with the physical activity performed outdoors [24,25,43].

In this study, we used the Leisure Satisfaction Scale (LSS) introduced by Beard and Ragheb (1980) [44] to determine the extent to which PMRP affects OAS. The researchers developed the scale through extensive theoretical inquiry and statistical tests based on various studies on leisure satisfaction [40,45,46]. The LSS categorizes six groups of 24 items, such as psychological, educational, social, resting, physical, and environmental factors, based on theoretical analysis. The LSS in OA may eventually change in a negative way due to environmental factors such as PM [27,28,29]. The outdoor activities satisfaction scale (OASS) was reconstructed combining LSS and the negative impact of PM (Table 3). It consists of four groups of 16 items, namely social, resting, physical, and environmental factors, related to PM.

OASS is directly related to PM. Furthermore, the PMRP is formed differently according to the health risks perceived by people. Ultimately, excessively formed risk perception by PM can lead to a reduction in the quality of OAS beyond simple worries. In South Korea, concerns that cause a PMRP are directly and indirectly related to the distrust of PM information and policies [15,17,19], the lack of evidence of health risks [1,47], and the source of public opinion (e.g., media or public participation) [14,15,20]. Therefore, this study further established the following three hypotheses based on previous studies:

 **Hypothesis 3 (H3).** 
*OAS has a positive effect on the level of interest in POP about PM.*


 **Hypothesis 4 (H4).** 
*OAS has a negative effect on the level of health risk perception about PM.*


 **Hypothesis 5 (H5).** 
*OAS has a positive effect on the level of trust in POP about PM.*


## 2. Materials and Methods

### 2.1. Research Model

This study aimed at investigating the structural relationships between PMRP and OAS in South Korea. In other words, it aims at establishing how risk perception related to PM affects the satisfaction of OA. PMRP consists of “interest level in POP”, “trust level in POP”, and “perception level of health risk” about PM. To achieve this, the following research model was designed (Figure 1).

### 2.2. Data Collection and Analysis

An Internet survey was conducted among Internet users. In 2017, 90.3% of Koreans were Internet users [48] and surveys concerning some psychological mechanisms, such as environmental perception, showed no significant difference between Internet users and non-users [49]. The survey was carried out from July 16 to August 16, 2019. During this period, people’s OA in South Korea was less affected by PM than at other times of the year. A total of 412 copies of the questionnaire were recovered (for details, see the Appendix A). The questionnaire consisted of five parts: respondents’ descriptive statistics, basic knowledge about PM, characteristics of OA related to PM, PMRP, and OAS. PMRP and OAS were measured on the basis of previous research using a 5-point Likert scale (1 point: very dissatisfied or disagree; 5 points: very satisfied or agree). To analyze these data, we used SPSS and AMOS 26.0. This analysis included structural equation modeling (SEM) and confirmatory factor analysis (CFA). SEM can control for measurement errors and use parameters to identify interdependencies [2,50]; therefore, it is appropriate to test this study model, which has the advantage of statistical evaluation by modeling causality among various variables.

## 3. Results

### 3.1. Sample Characteristics

Table 2 shows the characteristics of respondents who are aware that PM could affect their OA. The results show that there was almost no gender difference among the respondents, but the female ratio was slightly higher (217 women, 52.7%). The groups with a high interest in PM were located in Seoul (156, 37.9%) and Gyeonggi (129, 31.3%) province, were mostly aged 30–50 years old (267, 64.8%), and married (266, 64.6%). Vulnerable classes, as it relates to PM, were those in families with preschool children, those with respiratory and cardiopulmonary conditions, and the elderly (211, 54.6%).

The responses regarding basic knowledge of PM were as follows. Respondents who knew exactly about “the definition of PM” and “the conceptual difference between PM_10_ and PM_2.5_” accounted for 52.7% (217) and 45.3% (187), respectively. Those who found it hard to explain these two concepts were 45.6% (188) and 47.1% (194), respectively, which did not differ much from the number of respondents who knew exactly about such concepts. A percentage of 80.8% (333) of the respondents said that they became aware of PM in 2014 when the government began to deliver PM information by text message to the public through the FDFAS. This was also a time of explosive growth in PM articles on the Internet in South Korea [15,51]. The major sources of respondents’ information on PM were “Domestic Internet Sites and Applications (DISA)” and “TV/Radio”, at 68.7% (283) and 35.2% (145), respectively. This was followed by “International Internet Sites and Applications (IISA)”, accounting for 24.3% (100), due to a distrust in domestic information.

In the case of regular OA, cycling was the most reported activity (152, 36.9%) compared to other activities, such as walking. The four-level PM concentration standard (Air Quality Index [AQI]: very unhealthy, unhealthy, normal, and good) provided by the Korean government was used by most people as a reference (169, 41.0%), followed by the more stringent WHO standards (117, 28.4%) [52]. Health discomfort caused by PM was the highest for respiratory diseases (61.2%, 252), followed by eye diseases (52.2%, 215), skin diseases (33.0%, 136), and cardiovascular diseases (26.9%, 111). More than half of the respondents suffered from respiratory and eye diseases related to PM. In other words, about 90% of the respondents had diseases related to PM.

### 3.2. Results of the Confirmatory Factor Analysis

The Kaiser-Mayer-Olkin (KMO) test and Bartlett’s spherical test were used to determine whether the measures of PMRP and OAS variables were suitable for factor analysis. As a result, a KMO value of 0.767 was derived, and in Bartlett’s spherical test the null hypothesis was rejected with a significant probability of 0.000 under the significance level of 1%. Therefore, the suitability of factor analysis was confirmed. Variables of PMRP were derived from Kim and Lee’s (2019) [15] previous study on PMRP related to OA in South Korea. PMRP showed up in 12 factors. These factors were categorized into four dimensions: “interest level in POP”, “trust level in POP”, “level of PM health risk perception”, and “other”. Other factors, including “level of PM recognition” and “exposure level of PM information”, showed a low level of significance; therefore, we readjusted three parts of 10 factors (Table 3). To examine OAS variables based on Beard and Ragheb (1980) [44], the data were categorized into four factors, and the average value for each factor was used (Table 3). The Cronbach’s alpha value, which is an index indicating the consistency of measurement variables within each factor, ranged from 0.72 to 0.94. In the social sciences, values above 0.5 are reported to be significant [53].

A CFA was conducted to verify how well the measured variables of the model represent the latent variables. The results of the CFA are shown in Table 3. An appropriate cut-off-value of the ideal model fit has been presented in previous studies [2,54]. In this study, the goodness of fit statistics were found to be X^2^/DF = 1.949 *** (<3.0), GFI = 0.961, CFI =0.979, IFI = 0.979, AGFI = 0.933 (>0.9), and RMSEA = 0.048 (<0.5). Thus, this was found to be suitable for forming a structural model using the latent variables.

### 3.3. The Results of the Structural Equation Model

The results of the analysis of the effect of PMRP on OAS are shown in Table 4 and Figure 2. The path coefficients among the latent variables derived from the SEM are β values, which indicate interest level in POP, trust level in POP, health risk perception level, and OAS, respectively as −0.23 (*p* < 0.01), 0.56 (*p* < 0.001), −0.34 (*p* < 0.001). The level of interest in POP negatively affected the health risk perception level, whereas the trust level in POP affected it positively. As a result, health risk perception levels negatively affected OAS. Therefore, H1, H2, and H4 were supported. In other words, the more people are not interested in POP related to PM, the more they trust POP related to PM, and the higher the health risk perception level was related to PM. However, the higher the health risk perception level related to PM, the lower the OAS. This result is similar to previous studies on the negative aspects of OA caused by PMRP [27,28,29]. In addition, the path coefficient between interest level in POP related to PM and OAS was 0.17 (*p* < 0.001), which showed a significant positive impact; therefore, H3 was supported. This means that the higher the interest level in POP related to PM, the higher the OAS. Meanwhile, H5 had a positive impact, but the result was rejected because it was not statistically significant. Therefore, the research model in Figure 1 is now presented in a revised form in Figure 2. Lastly, the mediating effects of trust level and interest level in H4 measured as a direct effect of −0.34 and an indirect effect of −0.13, and a total effect of −0.47.

## 4. Discussion

This study analyzed how PMRP in Koreans affected their OAS. The main findings suggest that PMRP is higher when people are not interested in POP related to PM or do not trust it; the health risk perception level increased, but OAS decreased. The more people are interested in forming POP, the higher the OAS. Although the trust level of POP had a positive impact on OAS, it was rejected because the level was not statistically significant. In other words, when an individual’s interest is high and they do not react sensitively to the surrounding effects of PM information, it has a positive effect on PM health risk perception, which has a positive effect on OAS. This indicates that health risk perceptions are formed negatively as people are overexposed to fragmented media [35,39] or sensitive to frequent policy changes [17,19], in addition to environmental stressors, such as PM [55]. It is also consistent with the results of previous studies showing that the increasing PM health risk perception negatively affects OAS [26,32,55]. Furthermore, people who search for PM information directly, rather than passively sharing worries and concerns about PM, establish their OA standards and become more active. As a result, there is a positive correlation between OA and voluntary PM interest [15,56].

The novelty of this study stems from the fact that it evaluated OAS and discussed the results concerning people who have experienced PM (Table 1). In this survey, about 70% of respondents lived in Seoul and Gyeonggi provinces. These areas in South Korea are representative areas where the PM concentration has been serious [57]. Additionally, these areas are representative given the large population [58]; we aimed to conduct the survey with people who have experienced many problems related to PM. The PM fear in South Korea is high enough to be included as a social disaster, but the proportion of people who cannot distinguish between the PM definition and PM types accounted for about 50% of the respondents. This means that participants were found to form a PMRP without having acquired correct PM information [47]. The PM perception period was found to have an important connection with national policy implementation [17,59]. In some cases of distrust in domestic information, some people used foreign information as well to find ways to cope with PM. Although the national PM AQI was trusted by many, some of the respondents planned their OA around the more stringent WHO standards, while others responded to individual standards. Therefore, different ideas about standards cause different risk perceptions [15,59]. The health diseases caused by PM were found to affect the respiratory system, eyes, and skin in the short term. However, a long-term follow-up is necessary because different results are expected depending on the long-term impact of PM.

Although some risk perception studies on air pollution such as PM have been conducted, most of them used national data or averages of respondents’ descriptive statistics (e.g., gender, age, or income) [2,33]. Thus, discussion from a macro perspective is recommended. The risk perception approach of this study, however, is based on the premise that people have different ideas about risk [31,60]. If the environmental risk perception is different for each person, it is difficult to solve the risk problem without communicating at people’s various levels of understanding [61]. Hence, it is important to recognize the differences in the diversity and perceptions of the public, and not to depend only on the judgment of experts (or policymakers) or the unilateral dissemination of risk information [17,20]. In addition, understanding PM and communicating with experts can reduce fear and advance the debate on environmental risks. Therefore, to reduce PMRP and increase OAS, the media and the policymakers who form PM opinion at the public level [33,51] need to pay more attention and provide information on health risks related to PM along the lines recommended above.

This study has some limitations. First, there are a few previous studies on PMRP in developing countries, and it was difficult to discuss the PMRP situation in other countries. Second, the survey was conducted in summer to minimize the effects of PM and reduce seasonal variation. Thus, the potential results of a similar study conducted in winter or spring, when PM is more severe, were not considered in this study because they may distort the public’s thinking. However, since the characteristics of people’s thoughts about PM may change in the long term, follow-up studies on seasonal differences should be conducted. Third, the sample size (412 people) is limited for identifying relationships among PMRP and other variables such as age, residence, and OA type. Future research with a larger sample can better clarify how the public experiences PMRP.

In fact, the PM concentration in South Korea has been partially improved by the PM reduction strategies. However, PM concentration is still serious when compared to WHO standards and the levels of OECD countries; this must be solved in the future. Therefore, stakeholders should understand the population’s process of forming PMRP, and strategic policy formulation with phased implementation that reflects the public’s policy needs and the state’s current situation should be considered comprehensively. Furthermore, Korea’s government should present health risks at the level of those who experience PM rather than suggesting integrative social plans or government-led PM policies for the entire population. In addition, the media should help the public to understand PM by providing correct information, and risk communication skills will be needed by advisors to guide safe OA.

To sum up, many media and local governments in South Korea are providing information to the public on health risks without accurate education or promotion related to PM. However, despite the necessity of communicating about PM risk, incorrect information dissuades the public from distrusting PM data and policies. As a result, people’s quality of life is declining because of the fear of PM. In fact, there are many discussions about PMRP, such as fragmented media coverage, rapidly changing policies, and health risks, and this influences the public the most. From this perspective, the main finding is that perceptions of health risks have a direct impact on OAS. In addition, perceptions about the level of health risk can be shaped differently depending on the public interest and trust level in POP. Therefore, training and public relation processes are needed, along with experienced risk communication that can help the public understand PM and promote social consensus through staged discussions.

## 5. Conclusions

This study investigated the causal relationship between PMRP in the short term and OAS directly related to the health of those with a high perception of suffering from PM exposure. The health risk perception level related to PM was identified to be directly related to the interest and trust level in POP. Also, the level of health risk perception related to PM showed the greatest impact on OAS. As a result, it is very important that the findings derived through these processes are applied to future research classified by OA type so that people will not be disturbed by the fear of PM and will have a PMRP that guarantees safety. Therefore, these results are expected to contribute to risk communication guidelines in public opinion reporting and to the direction of environmental health policies in developing countries with high levels of air pollution, such as PM.

## Figures and Tables

**Figure 1 ijerph-17-01613-f001:**
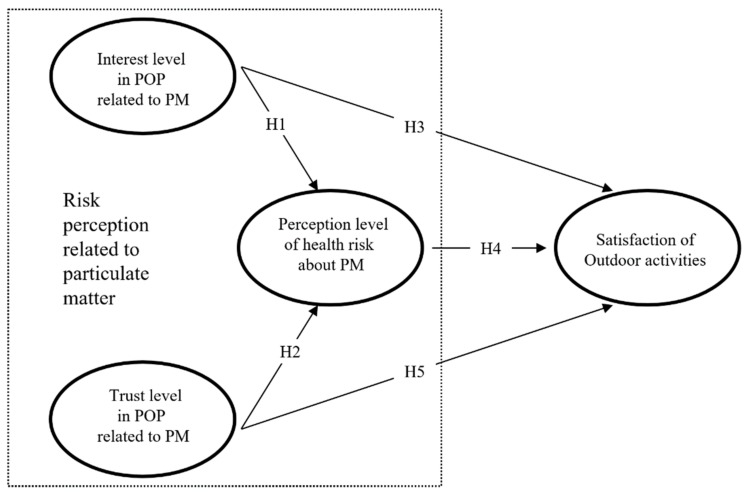
Research model. POP: public opinion or policy; PM: particulate matter; OAS: outdoor activity satisfaction.

**Figure 2 ijerph-17-01613-f002:**
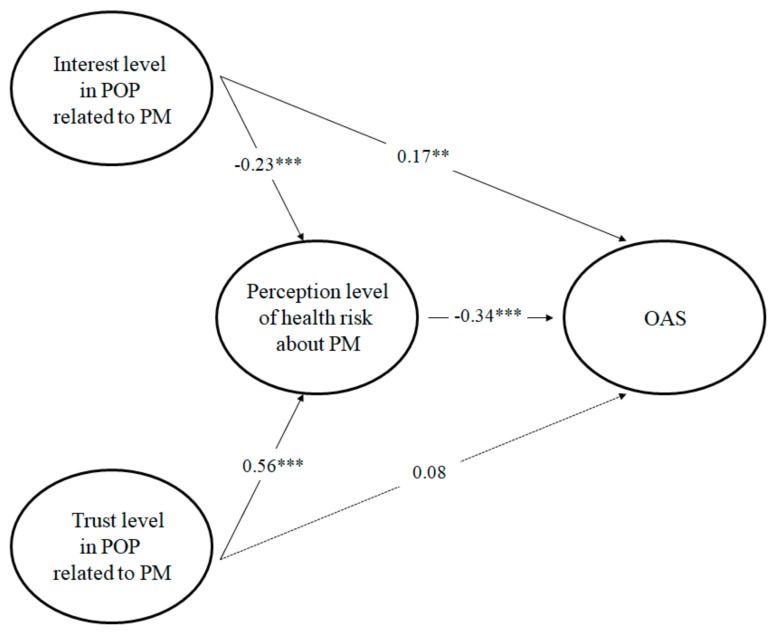
Results of the structural equation model.** *p* < 0.01, *** *p* < 0.001. X^2^/DF = 1.949 (*p* = 0.000), GFI = 0.961, CFI =0.979, IFI = 0.979, AGFI = 0.933, RMSEA = 0.048. POP: public opinion or policy; PM: particulate matter; OAS: outdoor activity satisfaction.

**Table 1 ijerph-17-01613-t001:** PM Criteria in South Korea.

Division (Unit: μg/m^3^)	Environmental Standards					AQI		FDFAS Standards *	
	1993	2001	2007	2011	2018	Unhealthy Level	Very Unhealthy Level	Watch Level	Warning Level
TSD	150/year	-	-	-	-	-	-	-	-
	300/day	-	-	-	-	-	-	-	-
PM10	80/year	70/year	50/year	50/year	50/year	80/h	150/h	150/h	300/h
	150/day	150/day	100/day	100/day	100/day				
PM2.5	-	-	-	25/year	15/year	35/h	75/h	35/h	75/h
	-	-	-	50/day	35/day				
WHO level		Interim Target 1	Interim Target 2		Interim Target 3	WHO recommendation criteria PM_10_ 20/year, 50/day PM_2.5_ 10/year, 25/day			

Note: 1. TSD: Total Suspended Particles, AQI: Air Quality Index, FDFAS: South Korea’s Fine Dust Forecasting and Alarming System 2. * The Korean government sends text messages about high-concentration PM when criteria of PM lasts over 2 hours [10].

**Table 2 ijerph-17-01613-t002:** Characteristics of respondents.

Division	Question	Response	Frequency	Percentage
	Gender	Male	195	47.3
		Female	217	52.7
	Residence	Seoul	156	37.9
		Gyeonggi	129	31.3
		Other areas	127	30.8
	Age	Up to 30	75	18.2
Descriptive statistics		31–40	143	34.7
		41–50	124	30.1
		51 or older	70	17.0
	Marital Status	Married	266	64.6
		Single	146	35.4
	Family member	Preschooler	124	26.8
	(vulnerable class	The elderly	66	14.3
	related to PM;	The sick	71	15.4
	multiple responses)	None	201	43.5
	Definition	Know exactly	217	52.7
	of PM	Hard to explain	188	45.6
		Don’t know	7	1.7
	Differences	Know exactly	187	45.4
	between	Hard to explain	194	47.1
	PM_10_ and PM_2.5_	Don’t know	31	7.5
Basic knowledge about PM	When awareness of PM started	Before 2014	79	19.2
		Since 2014	333	80.8
	Sources of	TV/ Radio	145	35.2
	PM information	DISA *	283	68.7
	(multiple	IISA **	100	24.3
	responses n = 698)	National text message	99	24.0
		Recommendations	60	14.8
		Others	11	2.7
	Regular OA	Cycling	152	36.9
		Other OAs	260	63.1
	PM	WHO	117	28.4
	concentration	Individual	60	14.6
	standard	Government	169	41.0
		Family/ friend	35	8.5
		Don’t care	31	7.5
Outdoor Activities related to PM	Health discomfort caused by PM	Eye Skin	215 136 252	52.2 33.0 61.2
	(multiple responses	Respiratory		
		Cardiopulmonary	111	26.9
	n = 761)	None	47	11.4
	OA action plan	Don’t care	44	10.7
	when PM concentration is at an unsafe level.	OA after using mask	118	28.6
		Cancellation of OA Change to indoor activities	84 166	20.4 40.3
Excluded	Missing		-	-
data ***	Non-participatory		-	-
	Total		412	100

Note: 1. DISA * Domestic Internet Sites and Applications IISA ** International Internet Sites and Applications 2. PM concentration at an unsafe level (unhealthy, very unhealthy) refers to more than PM_10_ 81 µg/m^3^ or PM_2.5_ 35 µg/m^3^ according to AQI. The atmospheric environment information measured by Korea’s Ministry of Environment was used [16]. 3. Excluded data *** The Internet survey was designed to not include missing or non-participatory data.

**Table 3 ijerph-17-01613-t003:** Results of the confirmatory factor analysis.

Questionnaire	Factor Loadings	t-Statistics	Cronbach’s α	CR	AVE
**PMRP**					
**Level of interest in POP related to PM**			0.72	0.95	0.85
Level of interest in public opinion related to PM	0.80	9.260 ***			
Prior to 2014, level of interest in policy related to PM	0.73	8.753 ***			
Since 2014, level of interest in policy related to PM	0.83				
**Level of trust in POP related to PM**			0.82	0.93	0.98
Reliability of information related to PM	0.79	9.271 ***			
Reliability of PM concentration	0.81	8.870 ***			
Reliability of domestic PM policies	0.82				
Reliability of international PM policies in Korea	0.67	12.837 ***			
**Level of health risk perception related to PM**			0.79	0.96	0.99
Experienced perception changes regarding PM concentration during OA	0.69				
Health risks of experienced PM_10_	0.86	10.962 ***			
Health risks of experienced PM_2.5_	0.84	10.932 ***			
**OAS (Despite PM concentration at an unsafe level.)**				0.96	0.99
**Resting satisfaction**			0.91		
My OA helps me to relax					
My OA helps to relieve stress					
My OA is good for emotional well-being					
I participate because of the outdoor fun					
**Social satisfaction**		22.311 ***	0.85		
My OA promotes social exchange					
My OA develops intimate relationships with others					
The people I met through my OA are friendly					
I hang out with people who enjoy their free time					
**Physical satisfaction**		21.359 ***	0.94		
My OA is a physical challenge					
My OA improves my fitness					
My OA helps me to recover physically					
MY OA helps me to stay healthy					
**Environmental satisfaction**		19.581 ***	0.90		
My OA place is comfortable					
My OA place is interesting					
My OA place is beautiful					
My OA place is well designed					

Note 1. *** *p* < 0.001. X^2^/DF = 1.949 (*p* =0.000), GFI = 0.961, CFI =0.979, IFI = 0.979, AGFI = 0.933, RMSEA = 0.048 2. PM concentration at an unsafe level means more than PM10 81 µg/m^3^ or PM2.5 µg/m^3.^

**Table 4 ijerph-17-01613-t004:** Results of the structural equation model.

Hypothesis Path				Support	β	*t*-Statistics	Direct Effects	Indirect Effects	Total Effects
H1	Level of POP participation	→	Level of health perception	Y	−0.23	−3.20 **	−0.23	-	−0.23
H2	Level of trust in POP	→	Level of health perception	Y	0.56	7.34 ***	0.56	-	0.56
H3	Level of POP participation	→	OAS	Y	0.17	4.09 *****	0.17	-	0.17
H4	Level of health perception	→	OAS	Y	−0.34	−5.08 ***	−0.34	−0.13	−0.47
H5	Level of trust in POP	→	OAS	N	0.08	1.39	0.08	-	0.08

** *p* < 0.01, *** *p* < 0.001, X^2^/DF = 1.949 (*p* = 0. 000), GFI = 0.961, CFI =0.979, IFI = 0.979, AGFI = 0.933, RMSEA = 0.048.

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
