# Peer review of "The Effects of Risk Perceptions Related to Particulate Matter on Outdoor Activity Satisfaction in South Korea"

_ijerph, 2020, doi:10.3390/ijerph17051613_

Round 1

Reviewer 1 Report

Review -Effect of Risk Perceptions Related to Particular Matter on Outdoor Activity Satisfaction in South Korea

First of all, I would like to congratulate the authors for completing this article which provide valuable inputs to the discussion on air pollution. There are, however, significant considerations and missing data that needs to be addressed prior to submission.

Start by putting your PM levels in perspective/context in order for readers to understand public risk perception. How high are levels and where are they measured? Sentence on page line 83 It is very hard to follow with all these abbreviations not typically used in my field; OA; OAS; PMRP. Make sure to have space btw last word and reference. You mention PM10, PM2.5 and PM but you should write out what their definition is i.e. PM10- PM of aerodynamic diameter smaller than 10µ The reason is that a high concentration of micro PM lasts a long time if it has occurred once, and people feel that the PM concentration is increasing, which affects their lives. This sentence is unclear and need to be restructured. Further it is unclear what the authors mean with micro PM? The authors should name a few of the PM reduction strategies than has been implemented and how successful they have been in order for readers to understand the problem. The authors write International Age Rating Coalition- do they really mean International Agency for Research on Cancer? The reason is that a high concentration of micro PM lasts a long time if it has occurred once, and people feel that the PM concentration is increasing, which affects their lives. Do the authors mean that micro (?) PM lasts a long time in the atmosphere? Unsure what they mean that PM is increasing without getting data on PM trends? Of course, the levels in South Korea probably do affect their health and wellbeing, are the authors suggesting it wouldn´t? As it is written now it implies that the public would be overreacting! Looking quickly at aqicn.org the levels in S Korea seems to exceed WHO guidelines by 5-9 times so public concerns seems to be in order. Korea’s PM problems are exacerbated by pollution from other countries such as Mongolia and China in winter and by domestic sources in the spring and fall[4]. This argument needs to be supported by scientific data not a meeting reference. It might be true but there is not enough scientific proof in the reference to show it. Further, with a quick look now (in winter time) levels in S Korea seems to range from 50-99 µg/m3 suggesting that at least half is generated nationally. This is not surprisingly given the high amount of diesel cars and coal powered energy. It is further suggested to be one of the most polluted countries in the world given that there are no really clean areas (below WHO guidelines) either in S Korea and contributing to 18 000 excess deaths per year (WHO data). It a study by Ministry of Environment in S Korea 97% of the population felt physical/psychological pain due to dust and most found the problem serious or very serious. It thus seems perfectly logical that people are worried about outdoor activities and it should be warranted that people use protective measures and perform physical activity in places with air filters. During physical activity the exposure to PM is exacerbated due to increased breathing patterns. If people are concerned about breathing dirty air, it is only countered by active policy measures. If these policy measures are not sufficient the public worry will increase of course. I would refrain to use words such as overreaction given that levels are 5-9 times the WHO AQG. PM is one of the major pollutants that cause air pollution in urban areas [2,3,11] PM is an air pollutant and thus not causing air pollution. The excluded group consisted of two cognitive questions on the level of PM information exposure and PM policies perception. Which ones were they and why were they excluded? These and all other questions must be in an additional file to the manuscript. The study requires more data on inclusion. Missing or non-participatory rates needs to be mentioned. We need to see how this question was put: exactly about “the definition of PM” and “the conceptual difference between PM10 and 163PM2.5”accounted for 52.7% (217) and 45.3% Discussion can not be evaluated until more data has been provided, regarding questionnaire and participatory rate and inclusion criteria, in order to draw conclusions from results.

Author Response

귀하의 의견에 감사드립니다.

그들은 우리가 원고의 질을 향상시키는 데 큰 도움이되었습니다.

귀하의 의견에 대한 답변은 첨부 파일에 있습니다.

Reviewer 2 Report

This paper reports a study of responses to an Internet survey of 412 people on how particulate matter risk perceptions affects outdoor activity satisfaction levels.  Overall, the paper is reasonably well written and the statistical analysis follows a conventional research protocol for analysis of the data.  Here are some comments to attend to in a revision. 

First, the causal ordering specified for the key variables of the structural equation model analysis (Figure 2) needs justification.  Why this causal ordering?  Why could OAS not affect the perception, interest, or trust variables?  In the text of the paper, you need to go into the prior research literature and the theoretical conceptualizations of how interest, perception, and trust levels are cognitively prior to psychological evaluations/satisfaction (with appropriate citations to prior theory and research).  You also could report estimates of statistical estimates of parameters, model fit, etc. that are consistent with the prior literature and that justify the causal ordering that you use.  Otherwise, the causal ordering of the structural equation model will appear to be arbitrary.

Second, you begin the second paragraph of the Discussion section with "The novelty of this study..."  Okay, that is fine.  There is another level of novelty that merits attention and that might be significant, namely:  Art there any substantively interesting and statistically significant interactions among the three prior (to OAS) variables of the structural equation model or of these variables with other variables (e.g., age of respondent) that merit comment?  The size of your sample (412) may not be sufficient to detect these, in which case you can mention this as an opportunity for further study.

Minor comment:  You use quite a few acronyms in the text.  Make sure these are defined at first usage; e.g., POP is used in the last sentence of the second paragraph of Section 1.1.1, but not defined until the next paragraph.

Author Response

(The authors gave the same response as above.)

Reviewer 3 Report

Overall an interesting and well presented study.  I offer minor suggestion below.

Line 19 & 21:  How was interest determined, and could surveying only interested persons, might this introduce some form of bias?

Line 37 & 38:  Is there a citation documenting that people feel that the PM concentration is increasing?

Line 48: Is there a citation that documents that OA is limited by PM?

Line 56: PM policies such as…?

Line 92: POP has yet to be defined.

Line 93 & 94: Does this sentence imply lack of internet elsewhere? 

Line 94: POP is defined here.

Line 137: How do we know these users are dissatisfied?  Why only choose dissatisfied individuals?

Table 1: Is your study demographic profile consistent with the demographic profile of Korea as a whole?

Line 260:  I don’t understand “communicating in a way that people can feel…”.

Author Response

(The authors gave the same response as above.)

Round 2

Reviewer 1 Report

Review -Effect of Risk Perceptions Related to Particular Matter on Outdoor Activity Satisfaction in South Korea

First of all, I would like to congratulate the authors for completing this article which provide valuable inputs to the discussion on air pollution. There are, however, still significant considerations and selection bias that needs to be addressed prior to submission.

  1. Start by putting your PM levels in perspective/context in order for readers to understand public risk perception. How high are levels and where are they measured?

Your answer: Response 1: Following your recommendation, Table 1 includes four stages reporting Korea’s fine dust forecasting and alarming system (FDFAS): good, normal, unhealthy, and very unhealthy. At an unhealthy and very unhealthy stage, the government recommends wearing a mask and refraining from outdoor activities. We used the criteria of the data measured by the Korean Ministry of Environment (www.airkorea.or.kr).

I can´t see that change-table 1 refer to respondent characteristics in the paper. If you mean the table inserted here, you should put it in the paper and discuss why there has not been any clear decline since 2012. See further down.

Remember that you should not refer to what the government think is unhealthy but WHO. Gov guidelines are often a trade off between industry and health.

I lack any PM levels as I asked for.

This is what you write in paper-As a result, the annual average concentration of PM10 in Seoul was 39 improved from 71 ㎍/㎥ (2001) to 49 ㎍/㎥ (2010), and an overall decrease has been shown until 2017 40 [7]. But looking at reference 7 there is no clear decrease between 2012 and 2017!

Thus, this paragraph is misleading and not scientifically correct if not backed with proof. The decrease in the last ten years is more relevant in this context as it can give people hope or despair and thus affect risk perception.

  1. Make sure to have space btw last word and reference.
  2. You mention PM10, PM2.5 and PM but you should write out what their definition is i.e. PM10- PM of aerodynamic diameter smaller than 10µ Please refrain from using dust when talking about PM.
  3. The authors should name a few of the PM reduction strategies than has been implemented and how successful they have been in order for readers to understand the problem.

Response 4: Following the reviewer’s suggestion, the introduction describes PM reduction strategies in South Korea.

Please make example of successful PM reductions schemes in the last ten years. To be clearer this could be banning of diesel cars from a certain year, reducing coal power plants and so on. Given the fact that the PM do not seem to decline since 2012 it is important for the reader to understand if measures have been enough.

  1. The reason is that a high concentration of micro PM lasts a long time if it has occurred once, and people feel that the PM concentration is increasing, which affects their lives. Do the authors mean that micro (?) PM lasts a long time in the atmosphere? Unsure what they mean that PM is increasing without getting data on PM trends? Of course, the levels in South Korea probably do affect their health and wellbeing, are the authors suggesting it wouldn´t? As it is written now it implies that the public would be overreacting! Looking quickly at aqicn.org the levels in S Korea seems to exceed WHO guidelines by 5-9 times so public concerns seems to be in order.

Thank you for deleting the overreaction statement. I am still not happy with the sentence as the definition on unhealthy air is not based on WHO but a governmental policy taking both industry and health into account. There is no reasons to believe that Koreans should be less vulnerable to PM exposure than people from Europe or US having lower levels regarded as unhealthy. If gov are not dealing with the problem efficiently (as can be seen by no decrease the last 5-6 years) public can distrust public data and rely on other media for risk perception.

  1. Korea’s PM problems are exacerbated by pollution from other countries such as Mongolia and China in winter and by domestic sources in the spring and fall[4]. This argument needs to be supported by scientific data not a meeting reference. It might be true but there is not enough scientific proof in the reference to show it. Further, with a quick look now (in winter time) levels in S Korea seems to range from 50-99 µg/m3 suggesting that at least half is generated nationally. This is not surprisingly given the high amount of diesel cars and coal powered energy. It is further suggested to be one of the most polluted countries in the world given that there are no really clean areas (below WHO guidelines) either in S Korea and contributing to 18 000 excess deaths per year (WHO data). It a study by Ministry of Environment in S Korea 97% of the population felt physical/psychological pain due to dust and most found the problem serious or very serious. It thus seems perfectly logical that people are worried about outdoor activities and it should be warranted that people use protective measures and perform physical activity in places with air filters. During physical activity the exposure to PM is exacerbated due to increased breathing patterns.

This has not been addressed properly.

  1. If people are concerned about breathing dirty air, it is only countered by active policy measures. If these policy measures are not sufficient the public worry will increase of course. I would refrain to use words such as overreaction given that levels are 5-9 times the WHO AQG.

This has not been addressed properly.

  1. The study requires more data on inclusion. Missing or non-participatory rates needs to be mentioned.

Selection bias of using the methods you describe must be clearly mentioned. Who are the people at the five Internet communities and how are they representable for the public as whole. Could it be some selection bias? Use discussion to expand on this.

  1. We need to see how this question was put: exactly about “the definition of PM” and “the conceptual difference between PM10 and 163PM2.5”accounted for 52.7% (217) and 45.3%

Thank you for the questions but a bit hard to judge if you do not put how precise the answer needed to be.

  1. Discussion can not be evaluated until more data has been provided, regarding questionnaire and participatory rate and inclusion criteria, in order to draw conclusions from results.

Author Response

우리의 논문에 대한 관심과 상세한 의견과 제안에 감사드립니다. 귀하의 의견에 대한 답변은 첨부 파일에 있습니다.

Reviewer 2 Report

The revisions made in this manuscript have been responsive to my comments on the previous version. 

Author Response

우리의 논문에 대한 관심과 상세한 의견과 제안에 감사드립니다. 귀하의 제안에 대한 답변을 통해 용지 품질이 향상되었습니다.
